# DEMYSTIFYING THE MYTHS AND LEGENDS OF NON-CONVEX CONVERGENCE OF SGD

## ABSTRACT

Stochastic gradient descent (SGD) and its variants are the main workhorses for solving large-scale optimization problems with nonconvex objective functions. Although the convergence of SGDs in the (strongly) convex case is well-understood, their convergence for nonconvex functions stands on weak mathematical foundations. Most existing studies on the nonconvex convergence of SGD show the complexity results based on either the minimum of the expected gradient norm or the functional sub-optimality gap (for functions with extra structural property) by searching the entire range of iterates. Hence the last iterations of SGDs do not necessarily maintain the same complexity guarantee. This paper shows that *an $\epsilon$-stationary point exists in the final iterates of SGDs,* given a large enough total iteration budget, $T$, not just anywhere in the entire range of iterates — a much stronger result than the existing one. Additionally, our analyses allow us to measure the *density of the $\epsilon$-stationary points* in the final iterates of SGD, and we recover the classical $O(\frac{1}{\sqrt{T}})$ asymptotic rate under various existing assumptions on the objective function and the bounds on the stochastic gradient. As a result of our analyses, we addressed certain myths and legends related to the nonconvex convergence of SGD and posed some thought-provoking questions that could set new directions for research.

## 1 INTRODUCTION

We consider the *empirical risk minimization* (ERM) problem:

$$\min_{x \in \mathbb{R}^d} \left[ F(x) := \frac{1}{n} \sum_{i=1}^{n} f_i(x) \right], \tag{1}$$

where $f_i(x) := \mathbb{E}_{z_i \sim \mathcal{D}_i} l(x; z_i)$ denotes the loss function evaluated on input, $z_i$, sampled from its distribution, $\mathcal{D}_i$. Additionally, let $F$ be nonconvex, lower bounded, with Lipschitz continuous gradient; see §3. ERM problems appear frequently in statistical estimation and machine learning, where the parameter, $x$, is estimated by the SGD updates (Bottou et al., 2018; Xu et al., 2021). For a sequence of iterates, $\{x_t\}_{t \geq 0}$ and a stepsize parameter, $\gamma_t > 0$, SGD updates are of the form:

$$x_{t+1} = x_t - \gamma_t g_t, \tag{2}$$

where $g_t$ is an unbiased estimator of $\nabla F(x_t)$, the gradient of $F$ at $x_t$; that is, $\mathbb{E}(g_t|x_t) = \nabla F(x_t)$. This approach, as given in (2), can be implemented by selecting an index $i(t)$ independently and uniformly from the set $[n]$, and processes $g_t = \nabla f_{i(t)}(x_t)$; the same index can be selected again.

The convergence of SGD for the strongly convex functions is well understood (Shalev-Shwartz et al., 2009; Gower et al., 2019; Shamir & Zhang, 2013), but its convergence of the *last* iterates for nonconvex functions remains an open problem. For a nonconvex function, $F$, the existing convergence analyses of SGD show, as $T \to \infty$, either (*i*) the minimum of the norm of the gradient function, $\min_{t \in [T]} \mathbb{E}\|\nabla F(x_t)\| \to 0$ (Ghadimi & Lan, 2013; Khaled & Richtárik, 2022; Stich & Karimireddy, 2020)[1] or (*ii*) the minimum sub-optimality gap, $\min_{t \in [T]} (\mathbb{E}(F(x_t)) - F_\star) \to 0$ (Gower et al., 2021;

---

[1] Some works show, as $T \to \infty$, the average of the expected gradient norm, $\frac{1}{T} \sum_t \mathbb{E}\|\nabla F(x_t)\| \to 0$ for fixed stepsize, or the weighted average of the expected gradient norm, $\frac{1}{\sum_t \gamma_t} \sum_t \gamma_t \mathbb{E}\|\nabla F(x_t)\| \to 0$ for variable stepsize (Bottou et al., 2018).

Lei et al., 2020). Notably, the first-class uses the classical $L$-smoothness, and the size of the gradient function, $\mathbb{E}\|\nabla F(x_t)\|$ to measure the convergence. Whereas the second class considers $F$ to have extra structural property, such as Polyak-Łojasiewicz (PL) condition (Gower et al., 2021; Lei et al., 2020) and the minimum sub-optimality gap is the measure of convergence. Nevertheless, in both cases the notion of $\epsilon$-stationary points[2] is weak as they only consider the minimum of the quantity $\mathbb{E}\|\nabla F(x_t)\|$ or $(\mathbb{E}(F(x_t)) - F_\star)$ approaching to 0 (as $T \to \infty$) by searching over the entire range of iterates, $[T]$. Alongside, by adding one more random sampling step at the end, $x_\tau \sim \{x_t\}_{t\in[T]}$, some works show $\mathbb{E}\|\nabla F_\tau(x_\tau)\| \to 0$ instead (Ghadimi & Lan, 2013; Stich & Karimireddy, 2020; Wang & Srebro, 2019).

In practice, we run SGD for $T$ iterations (in the order of millions for DNN training) and return the last iterate (Shalev-Shwartz et al., 2011). Therefore, we may ask: *How practical is the notion of an $\epsilon$-stationary point?* E.g., training ResNet-50 (He et al., 2016) on ImageNet dataset (Deng et al., 2009) requires roughly $600,000$ iterations. The present nonconvex convergence analysis of SGD tells only us that at one of those $600,000$ iterations, $\mathbb{E}\|\nabla F(x_t)\| \approx 0$. Indeed, this is not a sufficiently informative outcome. That is, the existing results treat all the iterations equally, do not motivate why we need to keep producing more iterations, but only reveal that as long we are generating more iterations, one of them will be $\epsilon$-stationary point. However, numerical experiments suggest that running SGD for more iterations will produce $\epsilon$-stationary points progressively more frequently, and the final iterates of SGD will surely contain more $\epsilon$-stationary points. Motivated by the remarkable fact that *the research community chooses high iteration counts to ensure convergence*, we investigate the theoretical foundations that justify the success of this common practice by asking the following questions: *Can we guarantee the existence of $\epsilon$-stationary point for SGD for the nonconvex case in the tail of the iterates, given a large enough iteration budget, $T$?* But guaranteeing the final iterates of SGD contain one of the $\epsilon$-stationary points alone does not conclude the task: we also would like to quantify the denseness of these $\epsilon$-stationary points among the last iterates. In all cases, in $T$ iterations, SGD achieves an optimal $O(\frac{1}{\sqrt{T}})$ asymptotic convergence rate for nonconvex, $L$-smooth functions (Carmon et al., 2020). A more refined analysis is required to capture this asymptotic rate.

We answer these questions and make the following contributions:

- For any stepsize (constant or decreasing), we show the existence of $\epsilon$-stationary points in the final iterates of SGD for nonconvex functions; see Theorem 2 in §4. We show that for every fixed $\eta \in (0, 1]$, there exists a $T$, large enough, such that there exists $\epsilon$-stationary point in the final $\eta T$ iterations. We can recover the classic $O(\frac{1}{\sqrt{T}})$ convergence rate of nonconvex SGD under no additional assumptions.
- An interesting consequence of our analyses is that we can measure the *concentration of the $\epsilon$-stationary points* in the final iterates of SGD—A first standalone result; see §4, Theorem 5 and 6. That is, we show that the concentration of the $\epsilon$-stationary points over the tail portion for the SGD iterates, $x_t$ is almost 1 for large $T$, where $t \in [(1 - \eta)T, T]$ and $\eta \in (0, 1]$. In optimization research community it is commonly agreed that without abundantly modifying the SGD algorithm, it is impossible to guarantee the convergence of the *last iterate* of SGD for nonconvex functions with the classic asymptotic rate (Dieuleveut et al., 2023, p.17). Therefore, on a higher level, our result clarifies common intuitions in the community: (*i*) why we need to keep on producing more iterations while running SGD; and (*ii*) why the practitioners usually pick the *last iterate of SGD* instead of randomly picking *one of the many iterates* as suggested by the present nonconvex theory. We acknowledge that these points are intuitive but taken for granted in the optimization research community without a theory. Additionally, we can extend our techniques to the convergence of random-reshuffling SGD (RR-SGD) and SGD for nonconvex and nonsmooth objectives; see §B.3 and §B.4.
- We informally addressed certain myths and legends that shape the practical success of SGD in the nonconvex world. We provide simple and interpretable reasons why some of these directions do not contribute to the success and posed some thought-provoking questions which could set new directions for research. We also support our theoretical results by performing numerical experiments on nonconvex functions, both smooth (logistic regres-

---

[2]A stationary point is either a local minimum, a local maximum, or a saddle point. In nonconvex convergence of SGD, $x$ is an $\epsilon$-stationary point if $\mathbb{E}\|\nabla F(x)\| \leq \epsilon$ or $(\mathbb{E}(F(x)) - F_\star) \leq \epsilon$.

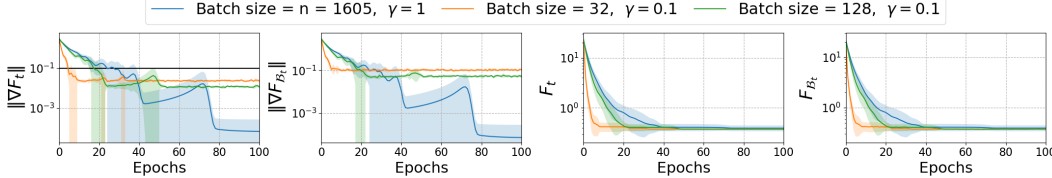

Figure 1: Average of 10 runs of SGD on logistic regression with nonconvex regularization. Batch size, $n = 1,605$, represents full batch. In the first column, the horizontal lines correspond to the precision, $\epsilon = 10^{-1}$, and conform to our theoretical result in Theorem 3—If the total number of iterations is large enough then almost all the iterates in the tail are $\epsilon$-stationary points.

> sion with nonconvex penalty) and nonsmooth (feed-forward neural network with ReLU activation); see §5. For publicly available code see §C.

## 2  BRIEF LITERATURE REVIEW

The convergence of SGD for convex and strongly convex functions is well understood; see §A for a few related work. We start with nonconvex convergence of SGD.

**Nonconvex convergence of SGD** was first proposed by Ghadimi & Lan (2013) for nonlinear, non-convex stochastic programming. Inspired by Nesterov (2003); Gratton et al. (2008), Ghadimi & Lan (2013) showed that SGD achieves $\min_{t\in[T]} \mathbb{E}\|\nabla F(x_t)\|^2 \le \epsilon$ after running for at most $O(\epsilon^{-2})$ steps—same complexity as the gradient descent for solving equation 1. Ghadimi et al. (2016) extended their results to a class of constrained stochastic composite optimization problems with loss function as the sum of a differentiable (possibly nonconvex) function and a non-differentiable, convex function. Recently, Vaswani et al. (2019) proposed a *strong growth condition* of the stochastic gradient, and showed that under SGC with a constant $\rho$, SGD with a constant step-size can attain the optimal rate, $O(\epsilon^{-1})$ for nonconvex functions; see Theorem 3 which is an improvement over Ghadimi & Lan (2013). Stich & Karimireddy (2020) proposed the *$(M, \sigma^2)$ noise bound* for stochastic gradients and proposed a convergence analysis for (compressed and/or) error-compensated SGD; see (Stich et al., 2018; Sahu et al., 2021). For nonconvex functions, Stich & Karimireddy (2020) showed $\mathbb{E}\|\nabla F(x_\tau)\| \to 0$, where $x_\tau \sim \{x_t\}_{t\in[T]}$. At about the same time, Khaled & Richtárik (2022) proposed a new assumption, *expected smoothness*, see Assumption 3, for modelling the second moment of the stochastic gradient and achieved the optimal $O(\epsilon^{-2})$ rate for SGD in finding stationary points for nonconvex, $L$-smooth functions. Among others, Lei et al. (2020) used Holder's continuity on gradients and showed the nonconvex convergence of SGD. Additionally, they showed the loss, $F$ converges to an *almost surely bounded random variable*. By using mini-batches to control the loss of iterates to non-attracted regions, Fehrman et al. (2020) proved the convergence of SGD to a minimum for *not necessarily* locally convex nor contracting objective functions. Additionally, for convergence of proximal stochastic gradient algorithms (with or without variance reduction) for nonconvex, nonsmooth finite-sum problems, see Reddi et al. (2016); Li & Li (2018); for non-convex problems with a non-smooth, non-convex regularizer, see Xu et al. (2019).

**Adaptive gradient methods** such as ADAM (Kingma & Ba, 2015), AMSGrad (Reddi et al., 2018), AdaGrad (Duchi et al., 2011) are extensively used for DNN training. Although the nonconvex convergence of these algorithms are more involved than SGD, they focus on the same quantities as SGD to show convergence. See nonconvex convergence of ADAM and AdaGrad by Défossez et al. (2020), nonconvex convergence for AdaGrad by Ward et al. (2019), Theorem 2.1; also, see Theorem 3 in (Zhou et al., 2020), and Theorem 2 in (Yang et al., 2016) for a unified analysis of stochastic momentum methods for nonconvex functions. Recently, Jin et al. (2022) proved almost sure asymptotic convergence of momentum SGD and AdaGrad.

**Compressed and distributed SGD** is widely studied to remedy the network bottleneck in bandwidth limited training of large DNN models, such as federated learning (Konečný et al., 2016; Kairouz et al., 2021). The convergence analyses of compressed and distributed SGD for nonconvex loss (Dutta et al., 2020; Xu et al., 2021; Alistarh et al., 2017; Sahu et al., 2021; Stich & Karimireddy, 2020) follow the same structure of the existing nonconvex convergence of SGD.

**Structured nonconvex convergence analysis of SGD and similar methods.** Gower et al. (2021) used extra structural assumptions on the nonconvex functions and showed SGD converges to a global

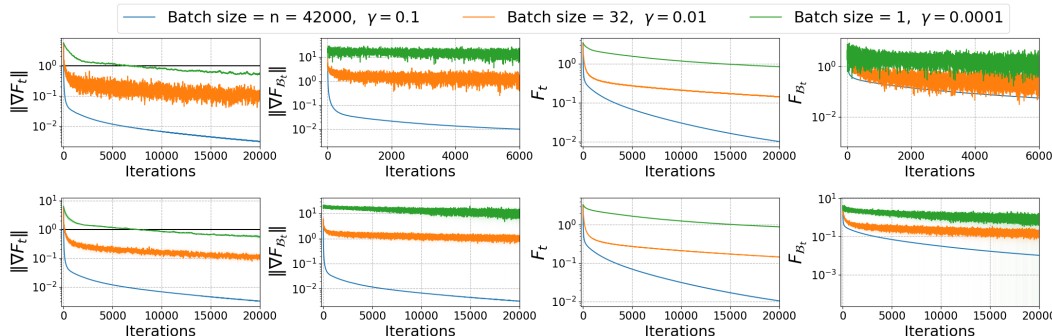

Figure 2: Performance of SGD on MNIST digit classification. The top row shows the result of 1 single run of SGD while the bottom row shows the result of the average of 10 runs. For the plots in the first column, the horizontal lines correspond to the precision, $\epsilon = 1$—For SGD, if the total number of iterations is large enough then the entire tail comprises of the $\epsilon$-stationary points. These plots confirm our observation in Figure 1 but for more complicated models.

minimum. Gorbunov et al. (2021) showed when $F$ is cocoercive (monotonic and $L$-Lipschitz gradient), the last-iterate for extra gradient (Korpelevich, 1976) and optimistic gradient method (Popov, 1980) converge at $O(\frac{1}{T})$ rate.

## 3 ASSUMPTIONS

**Assumption 1.** *(Global minimum) There exists $x_\star$ such that $F_\star := F(x_\star) \leq F(x)$ for all $x \in \mathbb{R}^d$.*

**Assumption 2.** *(Smoothness) For every $i \in [n]$, the function $f_i : \mathbb{R}^d \to \mathbb{R}$ is $L$-smooth, i.e. $f_i(y) \leq f_i(x) + \langle \nabla f_i(x), y - x \rangle + \frac{L}{2} \|y - x\|^2$ for all $x, y \in \mathbb{R}^d$.*

*Remark* 1. The above implies that $F$ is $L$-smooth.

**Bound on stochastic gradient.** There are different assumptions to bound the stochastic gradient. One may follow the model of Stich & Karimireddy (2020): Let the stochastic gradient in (2), $g_t$, at iteration $t$ be of the form, $g_t = \nabla F_t + \xi_t$, with $\mathbb{E}[\xi_t | x_t] = 0$. This leads to the $(M, \sigma^2)$-bounded noise assumption on the stochastic gradient (Stich & Karimireddy, 2020). Among different assumptions on bounding the stochastic gradient (Bottou et al., 2018; Ghadimi & Lan, 2013; Stich & Karimireddy, 2020; Lei et al., 2020; Gower et al., 2021; Vaswani et al., 2019; Dutta et al., 2020), recently, Khaled & Richtárik (2022) noted that the expected smoothness is the weakest among them and is as follows:

**Assumption 3.** *(Expected smoothness) There exist constants $A, B, C \geq 0$ such that for all $x_t \in \mathbb{R}^d$, we have $\mathbb{E}[\|g_t\|^2 \mid x_t] \leq 2A(F_t - F_\star) + B\|\nabla F(x_t)\|^2 + C$.*

Our analyses is based on Assumption 3 which contains other assumptions as special cases.

## 4 ANALYSIS

The standard analysis of nonconvex convergence of SGD use the minimum of the expected gradient norm, $\min_{t \in [T]} \mathbb{E}\|\nabla F(x_t)\| \to 0$, or the average of the expected gradient norm, $\frac{1}{T} \sum_t \mathbb{E}\|\nabla F(x_t)\| \to 0$ as $T \to \infty$, by using different conditions on the second moment of the stochastic gradient. In this section, we first modify that result and show that the last iterations of SGD would maintain the same complexity guarantee under different stepsize choices but under the same set of assumptions. Finally, an interesting consequence of our convergence analyses is that they allow us to measure the *density of the $\epsilon$-stationary points* in the final iterates of SGD without any additional assumptions. Denote $r_t = \mathbb{E}\|\nabla F(x_t)\|^2$, $\delta_t = \mathbb{E}(F_t) - F_\star$, and $D_T := \prod_{t=1}^{T}(1 + LA\gamma_t^2)$. Further denote the set of indices of $\epsilon$-stationary points, $S_\epsilon := \{t : r_t \leq \epsilon\}$. For $\eta \in (0, 1]$, let $S_{\epsilon,\eta} = S_\epsilon \cap [(1 - \eta)T, T]$. We start by quoting the general result for nonconvex convergence of SGD which is contained in the derivation of many known results in the literature.

**Theorem 1.** *Suppose Assumptions 1, 2, and 3 hold. With the notations above, we have for any learning rate $(\gamma_t)$ satisfying $\gamma_t \leq 1/LB$ that $\sum_{t=1}^{T} \gamma_t r_{t-1} \leq \left(2\delta_0 + \frac{C}{A}\right) D_T$.*

From Theorem 1, we obtain the classic nonconvex convergence of SGD with a constant learning rate. For decreasing learning rate of Theorem 1, see Corollary 2.

**Corollary 1** (Constant learning rate). *(Khaled & Richtárik, 2022) For constant stepsize $\gamma_t = \gamma$ in Theorem 1, we have* $\min_{1 \leq t \leq T} r_t \leq \frac{D_T}{\gamma T} \left( 2\delta_0 + \frac{C}{A} \right) \leq \frac{3\sqrt{LA}}{\ln(3)\sqrt{T}} \left( 2\delta_0 + \frac{C}{A} \right).$

Corollary 1 shows that in the entire range of iterates, $[T]$, there is one $r_t$ that approaches to 0 with a rate $O(\frac{1}{\sqrt{T}})$ as $T \to \infty$. But as we mentioned, this information is imprecise.

We now outline the proof of Theorem 1. Note that the key step using the $L$-smoothness of $F$ and *expected smoothness* of the stochastic gradients is the following inequality (Khaled & Richtárik, 2022, Lemma 2):

$$\gamma_t \left( 1 - \frac{LB\gamma_t}{2} \right) r_t \leq \left( 1 + LA\gamma_t^2 \right) \delta_t - \delta_{t+1} + \frac{LC\gamma_t^2}{2}. \tag{3}$$

Next, for any learning rate $(\gamma_t)$ chosen such that $\sum_{t=1}^{\infty} \gamma_t^2 < \infty$, we have $D_\infty \leq \exp(LA \sum_{t=1}^{\infty} \gamma_t^2) < \infty$. Unrolling the recurrence relation in (3) reduces the LHS to $\sum_{t=1}^{T} \prod_{i=t+1}^{T} (1 + LA\gamma_i^2)(\gamma_t - LB\gamma_t^2/2)r_{t-1}$. If we assume $\gamma_t \leq 1/LB$, then $\gamma_t - LB\gamma_t^2/2 \geq \gamma_t - \gamma_t/2 = \gamma_t/2$, and the lower bound of the LHS of the inequality (3) becomes $\frac{1}{2} \sum_{t=1}^{T} \gamma_t r_{t-1}$, as $\prod_{i=1}^{T} (1 + LA\gamma_i^2) \geq 1$. Hence, the proof of Theorem 1; see more detailed proof in §B.1.

Can we do any better? The answer to this question demystifies the first myth of the nonconvex convergence of SGD.

**Myth I: The nonconvex convergence of SGD is given by the minimum of the norm of the gradient function, $r_t$, over the range of entire iterates $[T]$.** For any $k \in [T]$, we can split the LHS $\frac{1}{2} \sum_{t=1}^{T} \prod_{i=t+1}^{T} (1 + LA\gamma_i^2)\gamma_t r_{t-1}$ of Theorem 1 into two sums as: $\frac{1}{2} \sum_{t=1}^{k-1} \prod_{i=t+1}^{T} (1 + LA\gamma_i^2)\gamma_t r_{t-1} + \frac{1}{2} \sum_{t=k}^{T} \prod_{i=t+1}^{T} (1 + LA\gamma_i^2)\gamma_t r_{t-1}$. Manipulating these quantities, we can obtain more precise information on the convergence of the $r_t$ in the tail portion of the iterates of SGD for nonconvex functions, which is given by $\min_{k \leq t \leq T} r_t$. To the best of our knowledge, this marks an improvement over the existing classical convergence results in Ghadimi & Lan (2013); Khaled & Richtárik (2022); Stich & Karimireddy (2020). We formally quote this in the following theorem; see proof in Appendix §B.

**Theorem 2.** *Suppose Assumptions 1, 2, and 3 hold. For any learning rate $(\gamma_t)$ satisfying $\gamma_t \leq 1/LB$, we have for any $k \in [T]$, $\min_{k \leq t \leq T} r_t \leq \left( 2\delta_0 + \frac{C}{A} - \frac{2\delta_T}{D_T} - \frac{1}{LA\gamma_1} \min_{1 \leq t \leq k-1} r_t \right) LA\gamma_k D_k \leq \left( 2\delta_0 + \frac{C}{A} \right) LA\gamma_k D_k.$*

In the subsequent analyses, we shed more light on the above result by considering different stepsizes.

**Constant step-size** $\gamma_t = \gamma$. Denote $D := (1 + L\gamma^2 A), E := \gamma(1 - LB\gamma/2), F := \frac{L\gamma^2 C}{2}$, and $W = \sum_{t=0}^{T} D^t$. By using (3), the existing results show, $\min_{t \in [T]} r_t \to 0$. First, we will improve this result.

Unrolling the recurrence in (3), and using the above mentioned notations, we have:

$$\sum_{t=0}^{T} (1 + L\gamma^2 A)^{T-t} r_t + \frac{2\delta_{T+1}}{\gamma(2 - L\gamma B)} \leq \frac{2\delta_0 (1 + L\gamma^2 A)^{T+1}}{\gamma(2 - LB\gamma)} + \frac{C[(1 + L\gamma^2 A)^{T+1} - 1]}{\gamma A(2 - LB\gamma)}. \tag{4}$$

Let $\eta \in (0, 1]$. The LHS in the inequality (4) is bounded from below by

$$\min_{(1-\eta)T \leq t \leq T} r_t \sum_{(1-\eta)T \leq t \leq T} (1 + L\gamma^2 A)^{T-t} \geq (\eta T - 1) \min_{(1-\eta)T \leq t \leq T} r_t;$$

if $LB\gamma \leq 1$ and $(1 + L\gamma^2 A)^{T+1} \leq 3$ then the RHS of (4) could be bounded from above by

$$\frac{6\delta_0}{\gamma} + \frac{2C}{\gamma A}. \tag{5}$$

Hence, we obtain

$$\min_{(1-\eta)T \leq t \leq T} r_t \leq 2 \left( 3\delta_0 + \frac{C}{A} \right) \frac{1}{(\eta T - 1)\gamma}. \tag{6}$$

Now, letting $\gamma := \sqrt{\frac{\ln 3}{(T+1)LA}}$, we can show the following result; see Appendix §B.2 for the proof.

**Theorem 3.** *Suppose Assumptions 1, 2, and 3 hold. Let $\epsilon > 0$ and $\eta \in (0, 1]$. If the number of iterations $T \geq 1$ satisfies $T \geq \max\left\{ \left( \frac{4\sqrt{2LA}(3\delta_0 + C/A)}{\varepsilon \eta \sqrt{\ln 3}} \right)^2, \frac{LB^2 \ln 3}{A} - 1, \frac{2}{\eta} \right\}$, then, there exists an index $t \geq (1 - \eta)T$ such that $\mathbb{E}\|\nabla F(x_t)\|^2 \leq \epsilon$.*

By controlling the stepsize parameter, Theorem 3 shows the $\epsilon$-stationary points exist in the final iterates of SGD for minimizing nonconvex functions.

*Remark 2.* Let $\epsilon > 0$. For choice of arbitrary $\eta \in (0, 1)$, and $T = \Omega\left(\max\{\frac{1}{\eta}, \frac{1}{\eta^2 \epsilon^2}\}\right)$, there exists a $t \in [(1 - \eta)T, T]$, such that, $\mathbb{E}\|\nabla F(x_t)\|^2 \leq \epsilon$. E.g., take $\eta = 0.05$ in the Theorem above. Then we know that the last 5% steps in the $T$ iterations will produce at least one $\varepsilon$-stationary point. For $\eta = 1$ in Theorem 3, we recover the classical asymptotic convergence rate of SGD, that is, $\min_{t \in [T]} \mathbb{E}\|\nabla F(x_t)\|^2 = O\left(\frac{1}{\sqrt{T}}\right)$.

*Remark 3.* Our choice of $\gamma$ makes the expression $(1 + L\gamma^2 A)^{T+1}$ contained in $[\sqrt{3}, 3]$ to the right of 1 on the real line. Any stepsize such that the expression is contained in an interval on the right of 1 will work, only with the difference in the constants in the estimations.

**Decreasing step-size.** We consider stepsize $\gamma_t = \frac{\gamma_0}{\sqrt{t+1}}$ with $\gamma_0 > 0$, and adopt a slightly different technique. Inspired by Stich & Karimireddy (2020), we define a non-negative, decreasing weighting sequence, $\{w_t\}_{t=0}^T$, such that $w_{-1} = 1$ and $w_t := \frac{w_{t-1}}{(1 + L\gamma_t^2 A)}$. Note that, the weights do not appear in the convergence result. With these weights, and by using the notations before, we can rewrite (3) as:

$$w_t \gamma_t (1 - \frac{LB\gamma_t}{2})r_t \leq w_t(1 + L\gamma_t^2 A)\delta_t - w_t \delta_{t+1} + \frac{w_t L\gamma_t^2 C}{2}.$$

Taking summation on above from $t = 0$ to $t = T$, we have

$$\sum_{t=0}^T w_t \gamma_t (1 - \frac{LB\gamma_t}{2})r_t \leq \delta_0 + \frac{LC}{2}\sum_{t=0}^T w_t \gamma_t^2. \tag{7}$$

The RHS of (7) is bounded above by

$$\delta_0 + \frac{LC}{2}\gamma_0^2(\ln(T+1) + 1). \tag{8}$$

Following the same technique as in the constant stepsize case, the LHS of (7) is bounded from below by

$$(1 - LA\gamma_0^2 \ln(T+1)) \min_{(1-\eta)T \leq t \leq T} r_t(\gamma_0(1 - \sqrt{1-\eta})\sqrt{T+1} - \frac{LB\gamma_0^2}{2}\ln(T+1)$$
$$+ \frac{LB\gamma_0^2}{2}\ln([(1-\eta)T] + 1)). \tag{9}$$

Combining (8) and (9), we can state the following Theorem; see Appendix §B.2 for the proof.

**Theorem 4.** *Suppose Assumptions 1, 2, and 3 hold. Let $\eta \in (0, 1]$. By choosing the stepsize $\gamma_t = \frac{\gamma_0}{\sqrt{t+1}}$ with $\gamma_0^2 < \frac{1}{LA \ln(T+1)}$, there exists a step $t \geq (1 - \eta)T$ such that $\mathbb{E}\|\nabla F(x_t)\|^2 \leq \frac{F_0 - F_\star + \frac{LC\gamma_0^2}{2}(\ln(T+1)+1)}{(1 - LA\gamma_0^2 \ln(T+1))\mathcal{C}(t)}$, where $\mathcal{C}(t) := (\gamma_0 \eta \sqrt{T+1} - \frac{LB\gamma_0^2}{2}\ln(T+1) + \frac{LB\gamma_0^2}{2}\ln([(1-\eta)T] + 1))$.*

*Remark 4.* From the previous Theorem we have for all fixed $\eta$ in $(0, 1]$ there exists a $t \in [(1 - \eta)T, T]$, such that $\mathbb{E}\|\nabla F(x_t)\|^2 = O\left(\frac{\ln(T+1)}{\sqrt{T+1}}\right)$. For $\eta = 1$, we recover the classical asymptotic convergence rate of SGD.

This result is also an improvement over the existing nonconvex convergence results of SGD for decreasing stepsize (Khaled & Richtárik, 2022; Stich & Karimireddy, 2020). See Corollary 3 in §B for decreasing stepsize of the form $\gamma_t = \gamma t^{-\alpha}$ with $\alpha \in (1/2, 1)$. One could think of other stepsizes in Theorem 2, e.g., $O(1/\sqrt{t})$, $O(1/\ln(t)\sqrt{t})$, and so on, and find the same guarantee.

> **Open questions**
>
> (*i*) For SGD, without an iteration budget, $T$, are there practical ways to detect the first iterate, $t$, for which $\mathbb{E}\|\nabla f(x_t)\|^2 \leq \epsilon$? (*ii*) How to terminate SGD other than the maximum number of iterations?

**Legend I: As long as (3) is used as a key descent inequality, there is no better convergence result.** Recall (3) is the key inequality to prove the convergence of SGD for nonconvex and $L$-smooth functions; see Theorem 1. We modified (3), and give the convergence of SGD in the tail for $\eta T$ iterations with $\eta \in (0, 1)$. For $\eta \in (0, 1)$, one could think of taking $k$ in specific form in Theorem 2, e.g., $k = (1 - \eta)T$, $k = (1 - 1/\eta)T$, $k = T - \eta \ln(T)$ or $k = T - \ln(T)$, and get a $O(\frac{1}{\sqrt{T}})$ convergence rate for the tail. However, if $k = T - \mathcal{C}$, where $\mathcal{C}$ is a constant, the rate disappears. Therefore, we must design a better descent inequality than (3).

**Myth II: Different stepsize choices result in novel convergence guarantees for SGD.** Several works propose a plethora of stepsize choices to boost the convergence of SGD (Gower et al., 2021; Loizou et al., 2021; Schaipp et al., 2023); (Wang et al., 2021) provide nonconvex convergence results for step decay. Since the quantity for convergence and the key descent step is problematic, different stepsize choices result only in incremental improvement other than a giant leap.

**Myth III: Better assumptions in upper bounding the stochastic gradients result in better convergence.** Recently, Allen-Zhu et al. (2019) showed why first-order methods such as SGD can find global minima on the training objective of multi-layer DNNs with ReLU activation and almost surely achieves close to zero training error in polynomial time. It was possible because the authors proved two instrumental properties of the loss, $F$: (*i*) gradient bounds for points that are sufficiently close to the random initialization, and (*ii*) semi-smoothness.

Assumptions on the bounds of the stochastic gradient are important, but this work does not judge which is better than the other as the literature has established potentially many interplays between them. Nevertheless, no significant improvements are possible following the present approach unless we propose completely new Assumptions, which should also be practical; but they are not easy to make. New assumptions on the upper bound of the stochastic gradients would only lead to minor improvements.

> **Open questions**
>
> Yu et al. (2021) assume the dissipativity property of the loss function, $F$ (less restrictive than convexity), and obtained guarantees on the last iterate of SGD. (*i*) What is the biggest class of nonconvex functions for which we can still have such guarantees? (*ii*) What are the minimal additional assumptions on $F$ to ensure the convergence of the last iterate?

**Legend II: If SGD is run for many iterates, then the tail will *almost surely* produce $\epsilon$-stationary points.** It is almost impossible to prove the last iterate convergence of SGD for nonconvex functions (see discussion in Dieuleveut et al. (2023)), but empirical evidence tells that the stationary points are mostly concentrated at the tail (Shalev-Shwartz et al., 2011). Nevertheless, how can we formalize it? The central idea in Theorem 3 is to bound a weighted sum of the gradient norms over all iterations from below by a partial sum over the last $\eta T$ iterations. Moreover, the result is not only mathematically stronger than the existing ones, but the simple trick of the partial sum over the last $\eta T$ iterations leads us to another significant result.

We know $S_{\epsilon, \eta} \neq \emptyset$ by Theorem 3. On one hand, we have

$$\sum_{t=(1-\eta)T}^{T} (1 + L\gamma^2 A)^{T-t} r_t > \sum_{\substack{t \in S_\epsilon^c \\ t \geq (1-\eta)T}} (1 + L\gamma^2 A)^{T-t} r_t > \sum_{t=(1-\eta)T+|S_{\epsilon,\eta}|}^{T} (1 + L\gamma^2 A)^{T-t} \epsilon$$

$$\geq \frac{(1 + L\gamma^2 A)^{\eta T - |S_{\epsilon,\eta}|} - 1}{L\gamma^2 A} \epsilon, \tag{10}$$

where $|S_{\epsilon,\eta}|$ denotes the cardinality of the set $S_{\epsilon,\eta}$. Note that, $\sum_{\substack{t \in S_\epsilon^c \\ t \geq (1-\eta)T}} (1 + L\gamma^2 A)^{T-t} r_t$ has $(\eta T - |S_{\epsilon,\eta}| + 1)$ terms; so, we lower bound them with the smallest of those many terms. On the

other hand, using (4) and (5) to bound the left hand side of (10), and rearranging the terms we obtain

$$(1 + L\gamma^2 A)^{\eta T - |S_{\epsilon,\eta}|} \leq \frac{6\delta_0 L\gamma A + 2CL\gamma}{\epsilon} + 1.$$

Taking logarithm to the previous inequality and rearranging the terms we get the following theorem:

**Theorem 5.** *Suppose Assumptions 1, 2, and 3 hold. Let $\epsilon > 0$ and $\eta \in (0, 1]$. For constant stepsize $\gamma_t = \gamma$, we have $\frac{|S_{\epsilon,\eta}|}{\eta T} \geq 1 - \frac{1}{T} \frac{\ln\left(\frac{6\delta_0 L\gamma A + 2CL\gamma}{\epsilon} + 1\right)}{\eta \ln(1 + L\gamma^2 A)}$.*

From the previous Theorem, we see that the density of the $\epsilon$-stationary points in the top $\eta$ portion of the tails approaches 1 as $T$ increases, which roughly speaking, tells us that for $T$ large enough almost all the iterations $x_t$ for $t \in [(1 - \eta)T, T]$ are $\epsilon$-stationary points.

Recall, from Theorem 3, if the total number of iterations, $T$ is large enough then there will be iterate, $t \in [(1 - \eta)T, T]$ to produce $\mathbb{E}\|\nabla F_t\|^2 \leq \epsilon$, where $\eta \in (0, 1]$. That is, Theorem 3 guarantees the existence of (at least one) stationary point(s) in the final iterates. However, one can argue that the effect of $\eta$ in the complexity can make it worse when $\eta$ is small, which requires the SGD to run for a sufficiently large number of iterations. Whereas the claim $\frac{|S_{\epsilon,\eta}|}{\eta T} \to 1$ as $T \to \infty$, says that running SGD for a large number of iterations, $T$ is not necessarily problematic, as almost surely the density of the stationary points in the tail portion will approach to 1, guaranteeing the entire tail comprising mostly of $\epsilon$-stationary points; see Figure 3.

Similarly, for the decreasing stepsize case, from Theorem 4, we have $S_{\epsilon,\eta} \neq \emptyset$. For $T$ large enough, we can lower bound the left side of the inequality (7) as

$$\sum_{t=0}^{T} w_t \gamma_t (1 - \frac{LB\gamma_t}{2}) r_t \geq \epsilon w_T \sum_{\substack{t \in S_\epsilon^c \\ t \geq (1-\eta)T}} \left(\gamma_t - \frac{LB\gamma_t^2}{2}\right) \geq \epsilon(1 + L\gamma_0^2 A(T+1))(\gamma_0\sqrt{T+1}$$

$$-\gamma_0\sqrt{(1 - \eta)T + |S_{\epsilon,\eta}|} - \frac{LB\gamma_0^2}{2}\ln(T+1)).$$

The above, combined with the upper bound in (8) can be written as

$$\gamma_0(\sqrt{T+1} - \sqrt{(1 - \eta)T + |S_{\epsilon,\eta}|}) \leq \underbrace{\frac{\delta_0 + \frac{LC}{2}\gamma_0^2(\ln(T+1) + 1)}{(1 + L\gamma_0^2 A(T+1))} + \frac{LB\gamma_0^2}{2}\ln(T+1)}_{:=\mathcal{D}},$$

which can be further reduced to

$$\frac{|S_{\epsilon,\eta}|}{\eta T} \geq 1 - 2\frac{\mathcal{D}}{\gamma_0\eta\sqrt{T}} + \frac{\mathcal{D}^2}{\gamma_0^2\eta T}. \tag{11}$$

Formally, we summarize it in the following theorem.

**Theorem 6.** *Suppose Assumptions 1, 2, and 3 hold. Let $\epsilon > 0$ and $\eta \in (0, 1]$. For decreasing step size, $\gamma_t = \frac{\gamma_0}{\sqrt{t+1}}$ with $\gamma_0 > 0$, we have $\frac{|S_{\epsilon,\eta}|}{\eta T} \geq 1 - O\left(\frac{1}{\sqrt{T}}\right)$.*

Similar to the argument for constant stepsize case, we conclude that the density of the $\epsilon$-stationary points in the top $\eta$ portion of the tail approaches to 1 as $T$ increases.

*Remark 5.* For stepsizes of the form $\gamma_t = \gamma t^{-\alpha}$ with $\alpha \in (1/2, 1)$, and $\gamma \in \mathbb{R}^+$, or other stepsizes, e.g., $O(1/\sqrt{t})$, $O(1/\ln(t)\sqrt{t})$, and so on, we can find similar guarantee.

**Connection with the high probability bounds.** The *high probability* convergence results, first used by Kakade & Tewari (2008), and then in Harvey et al. (2019a;b), proved convergence bounds for different class of loss functions (e.g., Lipschitz and strongly convex, but not necessarily differentiable; Lipschitz and convex, but not necessarily strongly convex or differentiable) by using rate Martingale and generalized Freeman's inequality. For Lipschitz and strongly convex functions, but not necessarily differentiable, Harvey et al. (2019a) proved that after $T$ steps of SGD, the error of the final iterate is $O(\log(T)/T)$ with high probability; also, see De Sa et al. (2015) for global convergence of SGD for matrix completion and related problems. Cutkosky & Mehta (2021) provided high-probability

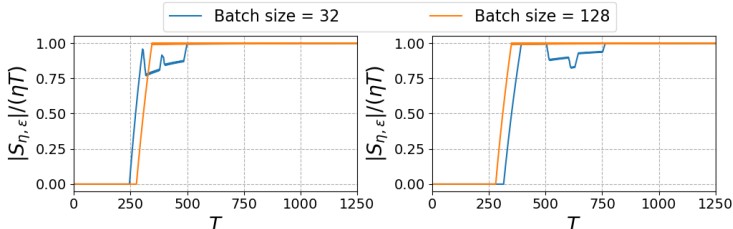

Figure 3: Concentration of the $\epsilon$-stationary points, $|S_{\epsilon,\eta}|/\eta T$ vs. Iterations for nonconvex logistic regression problem running SGD (left) and *random reshuffling SGD* (RR-SGD) (right) for choice of $\epsilon = 10^{-2}$ and $\eta = 0.2$. See theoretical convergence result of RR-SGD in Appendix B.3. As $T$ increases, $|S_{\epsilon,\eta}|/\eta T \to 1$ from below. The tail will *almost surely* produce $\epsilon$-stationary points and their density in the tail will approach to 1—It confirms our theoretical finding in Theorem 5.

bounds for nonconvex stochastic optimization with heavy tails. We refer to Arjevani et al. (2019) and references therein for the lower bound on the complexity of finding an $\epsilon$-stationary point using stochastic first-order methods. Harvey et al. (2019a) mentioned "high probability bounds are more difficult," as it controls the total noise of the noisy (sub)-gradients at each iteration, and not obvious that the error of the final iterate is tightly concentrated. To our knowledge, there is no trivial analysis. In contrast, we show the concentration of $\epsilon$-stationary point, $|S_{\epsilon,\eta}|/\eta T$ in the final iterates of nonconvex SGD as $T$ increases, approaches to 1, using simple arguments. Similar to Harvey et al. (2019a), our result can be generalized for SGD with suffix averaging or weighted averaging.

## 5 NUMERICAL EVIDENCE

We conduct experiments on nonconvex functions with $L$-smooth and non-smooth (for DNNs) loss to substantiate our theoretical results that are based on stochastic gradient, $g_t$. In practice, $g_t$ can be calculated by sampling and processing minibatches of data. Therefore, besides $\|\nabla F(x_t)\|$ and $F(x_t)$, we also track, the norm of the minibatch stochastic gradient, $\|\nabla F_{\mathcal{B}_t}\|$, and minibatch stochastic loss, $F_{\mathcal{B}_t}$. Note that, $\mathcal{B}_t$ is the selected minibatch of data at iteration $t$ and $F_{\mathcal{B}_t} := \frac{1}{|\mathcal{B}_t|} \sum_{i \in \mathcal{B}_t} f_i(x_t)$.

**Nonconvex and $L$-smooth loss.** We consider logistic regression with nonconvex regularization:

$$\min_{x \in \mathbb{R}^d} \left[ F(x) := \frac{1}{n} \sum_{i=1}^n \ln(1 + \exp(-a_i^\top x)) + \lambda \sum_{j=1}^d \frac{x_j^2}{1+x_j^2} \right],$$

where $a_1, a_2, ..., a_n \in \mathbb{R}^d$ are the given data, and $\lambda > 0$ is the regularization parameter. We run the experiments on the `ala` dataset from LIBSVM (Chang & Lin, 2011), where $n = 1605, d = 123$, and set $\lambda = 0.5$. Figures 1 (and Figure 4 in §B.3 for `RR-SGD`) shows the average of 10 runs of `SGD` with different minibatch sizes. The shaded area is given by $\pm\sigma$ where $\sigma$ is the standard deviation.

**Nonconvex and nonsmooth loss.** We use a feed forward neural network (FNN) for MNIST digit (LeCun et al., 1998) classification. The FNN has one hidden layer with 256 neurons activated by ReLU, and an 10 dimensional output layer activated by the softmax function. The loss function is the categorical cross entropy. We calculate the loss and the stochastic gradient during the training by using different minibatches. The entire loss and the full gradient are computed using all $n = 42 \times 10^3$ samples. For the average of 10 runs, the shaded area is given by $\pm\sigma$, where $\sigma \geq 0$ is the standard deviation, and $\gamma$ is the learning rate. Figures 2 in the main paper (and Figure 5 in §B.3) show if the total number of iterations is large enough then the entire tail comprising of the $\epsilon$-stationary points.

**Concentration of $\epsilon$-stationary points.** For $\epsilon = 10^{-2}$ and $\eta = 0.2$, in Figure 3, we plot the density of the $\epsilon$-stationary points, $|S_{\epsilon,\eta}|/\eta T$ as a function of iteration, $T$ for nonconvex logistic regression problems. As $T$ increases, $|S_{\epsilon,\eta}|/\eta T \to 1$ from below, and conform our result in §4.

## 6 CONCLUSION

To the best of our knowledge, this is the first work that proves the existence of *an $\epsilon$-stationary point in the final iterates of SGDs in optimizing nonconvex functions*, given a large enough total iteration budget, $T$, and *measures the concentration of such $\epsilon$-stationary points* in the final iterates. Alongside, by using simple reasons, we informally addressed certain myths and legends that determine the practical success of SGD in the nonconvex world and posed some new research questions.

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
