## A  BRIEF LITERATURE REVIEW—CONTINUED

**Convergence of structured convex and strongly convex functions.** Among the seminal works, Shamir & Zhang (2013) were the first to show that the final iterate has expected error $O(\frac{\ln T}{\sqrt{T}})$ for Lipschitz continuous functions and $O(\frac{\ln T}{T})$ for strongly convex functions. Jain et al. (2019) designed a new step size sequence for convex and strongly convex function to enjoy information theoretically optimal bounds on the suboptimality of last point of SGD as well as GD, with high probability. Harvey et al. (2019a;b) proved convergence bounds for non-smooth, strongly convex case by using Freeman's inequality.

**Stability and generalization bound**, that is, estimating the generalization error of learning algorithms is a classic problem and well studied; see for example, Rogers & Wagner (1978); Bousquet & Elisseeff (2002); Feldman & Vondrak (2018); Hardt et al. (2016); Lei & Ying (2020), among many. But stability and generalization of SGD is orthogonal to this work.

## B  PROOFS

### B.1  GENERAL CONVERGENCE ANALYSIS OF NONCONVEX SGD — CLASSICAL RESULT

The proofs provided in this section are the classic convergence analysis of nonconvex SGD, inspired and adapted from the existing literature; see Ghadimi & Lan (2013); Stich & Karimireddy (2020); Khaled & Richtárik (2022). We provide them for completeness.

*Proof of Theorem 1.* Recall the recursive relation (3) derived using the $L$-smoothness of $F$ and *expected smoothness* of the stochastic gradients $g_t$, e.g. see Lemma 2 in Khaled & Richtárik (2022)):

$$\gamma_t \left(1 - \frac{LB\gamma_t}{2}\right) r_t \le \left(1 + LA\gamma_t^2\right) \delta_t - \delta_{t+1} + \frac{LC\gamma_t^2}{2},$$

where $r_t := \mathbb{E}\|\nabla F_t\|^2$ and $\delta_t := \mathbb{E}F_t - F_\star$. Iterating the inequality above yields

$$\sum_{t=1}^{T} \prod_{i=t+1}^{T} \left(1 + LA\gamma_i^2\right) \left(\gamma_t - \frac{LB\gamma_t^2}{2}\right) r_{t-1} \le \prod_{t=1}^{T} \left(1 + LA\gamma_t^2\right) \delta_0 - \delta_T$$

$$+ \frac{LC}{2} \sum_{t=1}^{T} \prod_{i=t+1}^{T} \left(1 + LA\gamma_i^2\right) \gamma_t^2$$

$$= \left(\delta_0 + \frac{C}{2A}\right) \prod_{t=1}^{T} \left(1 + LA\gamma_t^2\right) - \delta_T$$

$$= \left(\delta_0 + \frac{C}{2A}\right) D_T - \delta_T \le \left(\delta_0 + \frac{C}{2A}\right) D_T,$$

where $D_T := \prod_{t=1}^{T} \left(1 + LA\gamma_t^2\right)$. Note, for any learning rate $(\gamma_t)$ chosen such that $\sum_{t=1}^{\infty} \gamma_t^2 < \infty$, then $D_\infty \le \exp(LA\sum_{t=1}^{\infty} \gamma_t^2) < \infty$. Next, if we assume $\gamma_t \le 1/LB$, then $\gamma_t - LB\gamma_t^2/2 \ge \gamma_t - \gamma_t^2/2 = \gamma_t/2$. This can be used to lower bound the LHS of the inequality above:

$$\sum_{t=1}^{T} \prod_{i=t+1}^{T} \left(1 + LA\gamma_i^2\right) \left(\gamma_t - \frac{LB\gamma_t^2}{2}\right) r_{t-1} \ge \frac{1}{2} \sum_{t=1}^{T} \prod_{i=t+1}^{T} \left(1 + LA\gamma_i^2\right) \gamma_t r_{t-1} \ge \frac{1}{2} \sum_{t=1}^{T} \gamma_t r_{t-1},$$
(12)

as $\prod_{i=1}^{T} \left(1 + LA\gamma_i^2\right) \ge 1$. $\qquad\square$

In particular, $\min_{1 \le t \le T} r_t \le \left(2\delta_0 + \frac{C}{A} - \frac{2\delta_T}{D_T}\right) \frac{D_T}{\sum_{t=1}^{T} \gamma_t} \le \left(2\delta_0 + \frac{C}{A}\right) \frac{D_T}{\sum_{t=1}^{T} \gamma_t}$. Note that $\min_{1 \le t \le T} r_t \to 0$ as $\sum_{t=1}^{\infty} \gamma_t = \infty$ and $\sum_{t=1}^{\infty} \gamma_t^2 < \infty$ (i.e. $D_\infty \le \exp(LA\sum_{t=1}^{\infty} \gamma_t^2) < \infty$ as $1 + x \le \exp(x)$).

The next corollary gives the nonconvex convergence of SGD for constant stepsize.

*Proof of Corollary 1.* First, we use that $\gamma \sum_{t=1}^{T} r_{t-1} \geq \gamma T \min_{1 \leq t \leq T} r_t$. The rest follows directly by letting $\gamma = \sqrt{\ln(3)/LAT}$ such that $D_T = (1 + LA\gamma^2)^T = (1 + \ln(3)/T)^T \leq e^{\ln(3)} = 3$; here we see $T$ as a fixed and known quantity. Note that the inequality $T \geq \ln(3)LB^2/A$ implies that $\gamma \leq 1/LB$ is satisfied. $\qquad\square$

The next corollary gives the classic nonconvex convergence of SGD for a special choice of decreasing stepsize.

**Corollary 2** (Decreasing learning rate of Theorem 1). *For decreasing learning rates on the form $\gamma_t = \gamma t^{-\alpha}$ with $\alpha \in (1/2, 1)$, we have*

$$\min_{1 \leq t \leq T} r_t \leq \frac{1-\alpha}{\gamma T^{1-\alpha}} \left( 2\delta_0 + \frac{C}{A} \right) \exp\left( \frac{2\alpha\gamma^2 LA}{2\alpha - 1} \right).$$

*Proof of Corollary 2.* Similarly to before, we first use that $\sum_{t=1}^{T} \gamma_t r_{t-1} \geq \min_{1 \leq t \leq T} r_t \sum_{t=1}^{T} \gamma_t$. The rest follows with the help of integral tests for convergence; $\sum_{t=1}^{T} t^{-\alpha} \geq \int_{1}^{T} x^{-\alpha}\,dx \geq T^{1-\alpha}/(1-\alpha)$ and $\sum_{t=1}^{T} t^{-2\alpha} = 1 + \sum_{t=2}^{T} t^{-2\alpha} \leq 1 + \int_{1}^{T} x^{-2\alpha}\,dx \leq 1 + 1/(2\alpha - 1) = 2\alpha/(2\alpha - 1)$, as $\alpha \in (1/2, 1)$. $\qquad\square$

### B.2 CONVERGENCE OF SGD USING EXPECTED SMOOTHNESS—BETTER RESULT FOR CONSTANT AND DECREASING STEPSIZE

We begin by giving the proof of Theorem 2.

*Proof of Theorem 2.* For any $k \in [T]$, we can lower bound the second term, $\frac{1}{2} \sum_{t=1}^{T} \prod_{i=t+1}^{T} \left(1 + LA\gamma_i^2\right) \gamma_t r_{t-1}$ of (12) in Theorem 1, as follows:

$$
\begin{aligned}
\frac{1}{2} \sum_{t=1}^{T} \prod_{i=t+1}^{T} \left(1 + LA\gamma_i^2\right) \gamma_t r_{t-1} =& \frac{1}{2} \sum_{t=1}^{k-1} \prod_{i=t+1}^{T} \left(1 + LA\gamma_i^2\right) \gamma_t r_{t-1} \\
&+ \frac{1}{2} \sum_{t=k}^{T} \prod_{i=t+1}^{T} \left(1 + LA\gamma_i^2\right) \gamma_t r_{t-1} \\
\geq& \frac{1}{2} \min_{1 \leq t \leq k-1} r_t \sum_{t=1}^{k-1} \prod_{i=t+1}^{T} \left(1 + LA\gamma_i^2\right) \gamma_t \\
&+ \frac{1}{2} \min_{k \leq t \leq T} r_t \sum_{t=k}^{T} \prod_{i=t+1}^{T} \left(1 + LA\gamma_i^2\right) \gamma_t \\
\geq& \frac{1}{2LA\gamma_1} \min_{1 \leq t \leq k-1} r_t \prod_{t=1}^{T} \left(1 + LA\gamma_t^2\right) \\
&+ \frac{1}{2LA\gamma_k} \min_{k \leq t \leq T} r_t \prod_{t=k}^{T} \left(1 + LA\gamma_t^2\right) \\
=& \frac{D_T}{2LA\gamma_1} \min_{1 \leq t \leq k-1} r_t + \frac{D_T}{2LA\gamma_k D_k} \min_{k \leq t \leq T} r_t \\
\geq& \frac{D_T}{2LA\gamma_k D_k} \min_{k \leq t \leq T} r_t.
\end{aligned}
$$

Note that, $\sum_{t=1}^{T} \prod_{i=t+1}^{T} \left(1 + LA\gamma_i^2\right) \left( \gamma_t - \frac{LB\gamma_t^2}{2} \right) r_{t-1}$ in (12) of Theorem 1 is bounded above by

$$\left( \delta_0 + \frac{C}{2A} \right) D_T.$$

Combining these together we obtain the result. $\qquad\square$

**Convergence of SGD for fixed stepsize—Proof of Theorem 3.** To help the reader, we sketched the key steps in the main paper. The technique used in our calculation is to estimate the quantities, $W, \frac{FW}{E}, \frac{D^{t+1}}{E}$, first, and then working on the upper and lower bounds of the quantities on the right and left side, respectively, of the inequality (6). The last estimates also require conditions on choosing the appropriate stepsize parameter. Below find our detailed derivations.

In the nonconvex convergence of SGD, by using the $L$-smoothness of $F$ and *expected smoothness* of the stochastic gradients, we arrive at the following key inequality; see Lemma 2 by Khaled & Richtárik (2022):

$$\gamma_t(1 - \frac{LB\gamma_t}{2})\mathbb{E}\|\nabla F_t\|^2 \leq (1 + L\gamma_t^2 A)(\mathbb{E}(F_t) - F_\star) - (\mathbb{E}(F_{t+1}) - F_\star) + \frac{L\gamma_t^2 C}{2}. \quad (13)$$

Denote $r_t = \mathbb{E}\|\nabla F_t\|^2, \delta_t = \mathbb{E}(F_t) - F_\star, D := (1 + L\gamma^2 A), E := \gamma(1 - \frac{LB\gamma}{2}), F := \frac{L\gamma^2 C}{2}$, and rewrite (13) as

$$\delta_{t+1} \leq D\delta_t - Er_t + F, \quad (14)$$

which after unrolling the recurrence becomes

$$\delta_{T+1} \leq D^{T+1}\delta_0 - E\sum_{t=0}^{T} D^{T-t}r_t + F\sum_{t=0}^{T} D^t. \quad (15)$$

Denote $W = \sum_{t=0}^{T} D^t$. Rearranging the terms again and dividing both sides by $E$ we have

$$\sum_{t=0}^{T} D^{T-t}r_t + \frac{\delta_{T+1}}{E} \leq \frac{D^{T+1}}{E}\delta_0 + \frac{FW}{E}. \quad (16)$$

Note that

$$W = \sum_{t=0}^{T} D^t = \frac{(1 + L\gamma^2 A)^{T+1} - 1}{L\gamma^2 A}, \quad \frac{FW}{E} = \frac{C[(1 + L\gamma^2 A)^{T+1} - 1]}{\gamma A(2 - LB\gamma)},$$

and

$$\frac{D^{T+1}}{E} = \frac{2(1 + L\gamma^2 A)^{T+1}}{\gamma(2 - LB\gamma)}.$$

Therefore, (16) can be written as

$$\sum_{t=0}^{T} (1 + L\gamma^2 A)^{T-t}r_t + \frac{2\delta_{T+1}}{\gamma(2 - L\gamma B)} \leq \frac{2(1 + L\gamma^2 A)^{T+1}}{\gamma(2 - LB\gamma)}\delta_0 + \frac{C[(1 + L\gamma^2 A)^{T+1} - 1]}{\gamma A(2 - LB\gamma)}. \quad (17)$$

Let $\eta \in (0, 1]$. Then the left-hand side in the inequality (17) is bounded from below by

$$\min_{(1-\eta)T \leq t \leq T} r_t \sum_{(1-\eta)T \leq t \leq T} (1 + L\gamma^2 A)^{T-t} \geq (\eta T - 1) \min_{(1-\eta)T \leq t \leq T} r_t;$$

if $LB\gamma \leq 1$ and $(1 + L\gamma^2 A)^{T+1} \leq 3$ then the right-hand side of (17) could be bounded from above by

$$\frac{6\delta_0}{\gamma} + \frac{2C}{\gamma A}. \quad (18)$$

Hence, we obtain

$$\min_{(1-\eta)T \leq t \leq T} r_t \leq 2\left(3\delta_0 + \frac{C}{A}\right)\frac{1}{(\eta T - 1)\gamma}. \quad (19)$$

Now, letting $\gamma := \sqrt{\frac{\ln 3}{(T+1)LA}}$, we are able to show the following result:

Let $T$ satisfy the inequality in the Theorem. We first estimate $(\eta T - 1)\gamma$ from below. We have

$$(\eta T - 1)\gamma = \frac{\eta T - 1}{\sqrt{T+1}}\frac{\sqrt{\ln 3}}{\sqrt{LA}} > \frac{\frac{1}{2}\eta T}{\sqrt{2T}}\frac{\sqrt{\ln 3}}{\sqrt{LA}} = \frac{\eta\sqrt{\ln 3}}{2\sqrt{2LA}}\sqrt{T}$$

since $T \geq 2/\eta$ and $T \geq 1$. Using this estimate in inequality (6), we get

$$\min_{(1-\eta)T \leq t \leq T} r_t \leq 2 \left( 3\delta_0 + \frac{C}{A} \right) \frac{2\sqrt{2LA}}{\eta \sqrt{\ln 3} \sqrt{T}}$$

which is less than $\varepsilon$ by the choice of $T$. Next, using the well-known inequality $1 + x \leq e^x$ for all $x \geq 0$, we have

$$(1 + L\gamma^2 A)^{T+1} = \left( 1 + \frac{\ln 3}{T+1} \right)^{T+1} \leq e^{\ln 3} = 3.$$

Finally, note that the inequality $T + 1 \geq LB^2 \ln 3/A$ implies that $LB\gamma \leq 1$ is satisfied. So, we verified that if $T$ is chosen as in the Theorem, all the needed inequalities for obtaining inequality (6) are true. This completes our proof of Theorem 3.

**Convergence of SGD for decreasing stepsize—Proof of Theorem 4.** We provide more details for the proof sketched in Section 3. Recall that $w_{-1} = 1$ and $w_t = \frac{w_{t-1}}{(1+L\gamma_t^2 A)}$ and recall (7)

$$\sum_{t=0}^{T} w_t \gamma_t (1 - \frac{LB\gamma_t}{2}) r_t \leq \delta_0 + \frac{LC}{2} \sum_{t=0}^{T} w_t \gamma_t^2. \tag{20}$$

Since $\{w_t\}_{t=0}^{T}$ is a non-negative, decreasing weighting sequence, we have $w_T \leq w_t \leq w_{-1} = 1$ for all $t \in [T]$. Now consider, $\gamma_t = \frac{\gamma_0}{\sqrt{t+1}}$ with $\gamma_0 > 0$ a decreasing stepsize sequence for all $t \in [T]$. As a consequence, we have

$$\sum_{t=0}^{T} w_t \gamma_t^2 \leq \gamma_0^2 \int_{t=0}^{T} \frac{1}{t+1} dt = \gamma_0^2 \left( \ln(T+1) + 1 \right).$$

Hence, the right hand side of (20) is bounded above by

$$\delta_0 + \frac{LC}{2} \sum_{t=0}^{T} w_t \gamma_t^2 \leq \left( \delta_0 + \frac{LC}{2} \gamma_0^2 \ln(T+1) \right). \tag{21}$$

Following the same technique as in the constant stepsize case, the left hand side of (20) is bounded from below by

$$\sum_{t=0}^{T} w_t \gamma_t \left( 1 - \frac{LB\gamma_t}{2} \right) r_t \geq w_T \min_{(1-\eta)T \leq t \leq T} r_t \left( \sum_{t=(1-\eta)T}^{T} \gamma_t \left( 1 - \frac{LB\gamma_t}{2} \right) \right). \tag{22}$$

We will pause here and estimate the quantities. It is straight-forward to find out

$$\sum_{t=(1-\eta)T}^{T} \gamma_t \geq \gamma_0 \int_{t=(1-\eta)T}^{T} \frac{1}{\sqrt{t+1}} dt = \gamma_0 (1 - \sqrt{1-\eta})\sqrt{T+1}.$$

and

$$\sum_{t=(1-\eta)T}^{T} \frac{LB}{2} \gamma_t^2 \leq \frac{LB}{2} \gamma_0^2 \int_{t=[(1-\eta)T]}^{T} \frac{1}{t+1} dt + 1 = \frac{LB\gamma_0^2}{2} \left( \ln(T+1) - \ln([(1-\eta)T]+1) \right).$$

Recall that, by definition

$$w_T = \frac{w_{-1}}{\Pi_{t=0}^{T}(1+LA\gamma_t^2)} \geq \frac{w_{-1}}{\left( \frac{\sum_{t=0}^{T}(1+LA\gamma_t^2)}{T+1} \right)^{T+1}}.$$

The last inequality is due to the arithmetic-geometric inequality.

Next, we upper bound

$$\left( \frac{\sum_{t=0}^{T}(1+LA\gamma_t^2)}{T+1} \right)^{T+1} \overset{\gamma_t = \frac{\gamma_0}{\sqrt{t+1}}}{=} \left( 1 + \frac{LA \sum_{t=0}^{T} \frac{\gamma_0^2}{t+1}}{T+1} \right)^{T+1} \leq \left( 1 + \frac{LA\gamma_0^2 \ln(T+1)}{T+1} \right)^{T+1}.$$

Considering $LA\gamma_0^2 \ln(T+1) < 1$ we have

$$\left(1 + \frac{LA\gamma_0^2 \ln(T+1)}{T+1}\right)^{T+1} \leq \frac{1}{1 - LA\gamma_0^2 \ln(T+1)}.$$

Taken together, we can find (22) is further lower bounded by

$$(1 - LA\gamma_0^2 \ln(T+1)) \min_{(1-\eta)T \leq t \leq T} r_t(\gamma_0(1 - \sqrt{1-\eta})\sqrt{T+1} - \frac{LB\gamma_0^2}{2}\ln(T+1)$$
$$+ \frac{LB\gamma_0^2}{2}\ln([(1-\eta)T]+1)). \tag{23}$$

Combining (21) and (23) completes the proof of Theorem 4 for stepsize $\gamma_t = \frac{\gamma_0}{\sqrt{t+1}}$.

Additionally, we show the nonconvex convergence of SGD for decreasing learning rates of the form $\gamma_t = \gamma t^{-\alpha}$ with $\alpha \in (1/2, 1)$ in the following Corollary.

**Corollary 3** (Decreasing learning rate of Theorem 2). *For decreasing learning rates on the form $\gamma_t = \gamma t^{-\alpha}$ with $\alpha \in (1/2, 1)$, we have after*

$$k \geq \left(\frac{1-\alpha}{\gamma\epsilon}\left(2\delta_0 + \frac{C}{A}\right)\exp\left(\frac{2\alpha\gamma^2 LA}{2\alpha - 1}\right)\right)^{\frac{1}{1-\alpha}}$$

*iterations that*

$$\min_{k \leq t \leq T} r_t \leq \epsilon, \quad \epsilon > 0.$$

*Proof of Corollary 3.* This results follows directly from Theorem 2 with use of Corollary 2, e.g. see the proof technique in Needell et al. (2014). $\square$

### B.3 CONVERGENCE OF RANDOM RESHUFFLING-SGD (RR-SGD)

The existing programming interfaces in ML toolkits such as PyTorch (Pytorch.org, 2019) and TensorFlow (tensorflow.org, 2015) use a different approach—*random reshuffling* or *randomness without replacement* (Mishchenko et al., 2020; Gürbüzbalaban et al., 2021). In this case, at each cycle, $t$, a random permutation, $\sigma_t$, of the set $[n]$ is selected, and one complete run is performed taking all indices from $\sigma_t$, which guarantees that each function in (1) contributes exactly once. Formally, RR-SGD updates are of the form:

$$x_{(t-1)n+i} = x_{(t-1)n+i-1} - \gamma_t g_{\sigma_t(i)}(x_{(t-1)n+i-1}), \quad i = 1, 2, ..., n; \ t = 1, 2, 3, ...,$$

where $g_{\sigma_t(i)}(x_j)$ is the stochastic gradient calculated at $x_j$. RR-SGD posses faster convergence than regular SGD (Mishchenko et al., 2020; Gürbüzbalaban et al., 2021), leaves less stress on the memory (cf. Section 19.2.1 in Bengio (2012)), and hence more practical.

We state the bounded variance assumption of gradients from Mishchenko et al. (2020) that is used in proving the nonconvex descent lemma of RR-SGD; see Lemma 1. For details of how Assumptions 3 and 4 are connected see Mishchenko et al. (2020).

**Assumption 4.** *(**Bounded variance of gradients**) There exist constants, $\mathcal{A}, \mathcal{B} \geq 0$, such that, for all $x \in \mathbb{R}^d$, the variance of gradients follow*

$$\frac{1}{n}\sum_{i \in [n]} \|\nabla f_i(x) - \nabla F(x)\|^2 \leq 2\mathcal{A}(F(x) - F_\star) + \mathcal{B}.$$

Recently, Mishchenko et al. (2020) showed a better nonconvex convergence of RR-SGD compared to prior work of Nguyen et al. (2021) without the bounded gradient assumption. Mishchenko et al. (2020) followed Assumption 4—bounded variance of gradients. In this section, we sketch the key steps of the convergence of RR-SGD. We start by quoting the key descent Lemma used for the convergence of RR-SGD from Mishchenko et al. (2020). We focus on constant stepsize case, results for decreasing stepsize follow the similar arguments.

**Lemma 1.** *Let $F$ follow Assumptions 1, 2, and 4, and the update rule in (24) is run for $T$ epochs. Then for $\gamma \leq \frac{1}{2Ln}$ and $t \in \{0, 1, \cdots T - 1\}$, the iterates of (24) satisfy*

$$(\mathbb{E}(F_{t+1}) - F_\star) \leq (1 + \mathcal{A}L^2 n^2 \gamma^3)(\mathbb{E}(F_t) - F_\star) - \frac{\gamma n}{2}(1 - \gamma^2 L^2 n^2)\mathbb{E}\|\nabla F_t\|^2 + \frac{L^2\gamma^3 n^2 \mathcal{B}}{2}, \tag{24}$$

*where $T$ denotes the total number of epochs.*

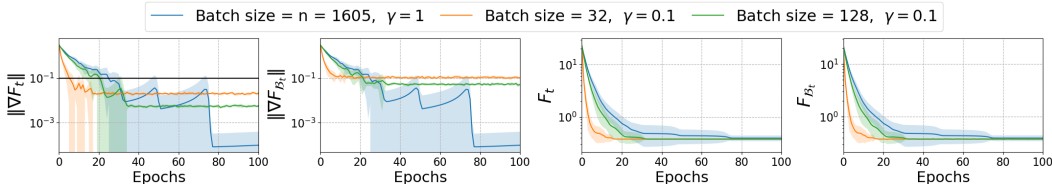

Figure 4: Average of 10 runs of `RR-SGD` on logistic regression with nonconvex regularization. Batch size, $n = 1605$, represents full batch. In the first column, the horizontal lines correspond to the precision, $\epsilon = 10^{-1}$, and conform to our theoretical result in Theorem 7—If the total number of iterations is large enough then almost all the iterates in the tail are $\epsilon$-stationary points.

Proceeding similarly as before, and letting $\gamma_t = \gamma := \left( \frac{\ln 3}{(T+1)\mathcal{A}L^2n^2} \right)^{\frac{1}{3}}$, we can show the following result.

**Theorem 7.** *Let $F$ follow Assumptions 1, 2, and 4, and the update rule in (24) is run for $T$ epochs. Let $\epsilon > 0$ and $\eta \in (0, 1]$. If the number of epochs $T > 1$ satisfies*

$$T \geq \max \left\{ 27 \left( 3\delta_0 + \frac{\mathcal{B}}{\mathcal{A}} \right)^3 \frac{\mathcal{A}L^2}{n\eta^2\varepsilon^2}, \frac{8Ln\ln 3}{\mathcal{A}} - 1, \frac{2}{\eta} \right\},$$

*then, there exists an index, $t \geq (1 - \eta)T$, such that $\mathbb{E}\|\nabla F_t\|^2 \leq \epsilon$.*

Denote $r_t = \mathbb{E}\|\nabla F_t\|^2$, $\delta_t = \mathbb{E}(F_t) - F_\star$, $D_2 := (1 + \mathcal{A}L^2n^2\gamma^3)$, $E_2 = \frac{\gamma n}{2}(1 - \gamma^2 L^2 n^2)$, $F_2 := \frac{L^2\gamma^3 n^2 \mathcal{B}}{2}$. Rewrite (24) as

$$\delta_{t+1} \quad \leq \quad D_2\delta_t - E_2 r_t + F_2,$$

which after unrolling the recurrence becomes

$$\delta_{t+1} \quad \leq \quad D_2^{t+1}\delta_0 - E_2 \sum_{j=0}^{t} D_2^{t-j}r_j + F_2 \sum_{j=0}^{t} D_2^j. \tag{25}$$

Denote $W_2 = \sum_{j=0}^{t} D_2^j$. Rearranging the terms again and dividing both sides by $E_2$ we have

$$\sum_{j=0}^{t} D_2^{t-j}r_j + \frac{\delta_{t+1}}{E_2} \quad \leq \quad \frac{D_2^{t+1}}{E_2}\delta_0 + \frac{F_2 W_2}{E_2}. \tag{26}$$

Note that

$$W_2 = \sum_{j=0}^{t} D_2^j = \frac{D_2^{t+1} - 1}{D_2 - 1} = \frac{(1 + \mathcal{A}L^2n^2\gamma^3)^{t+1} - 1}{\mathcal{A}L^2n^2\gamma^3},$$

$$\frac{F_2 W_2}{E_2} = \frac{L^2\gamma^3 n^2 \mathcal{B}}{2} \cdot \frac{(1 + \mathcal{A}L^2n^2\gamma^3)^{t+1} - 1}{\mathcal{A}L^2n^2\gamma^3} \cdot \frac{2}{\gamma n(1 - \gamma^2 L^2 n^2)} = \frac{\mathcal{B}[(1 + \mathcal{A}L^2n^2\gamma^3)^{t+1} - 1]}{\mathcal{A}\gamma n(1 - \gamma^2 L^2 n^2)},$$

and

$$\frac{D_2^{t+1}}{E_2} = \frac{2(1 + \mathcal{A}L^2n^2\gamma^3)^{t+1}}{\gamma n(1 - \gamma^2 L^2 n^2)}.$$

Therefore, (26) can be written as

$$\sum_{j=0}^{t} (1 + \mathcal{A}L^2n^2\gamma^3)^{t-j}r_j + \frac{2\delta_{t+1}}{\gamma n(1 - \gamma^2 L^2 n^2)} \quad \leq \quad \frac{2(1+\mathcal{A}L^2n^2\gamma^3)^{t+1}}{\gamma n(1-\gamma^2 L^2 n^2)}\delta_0 + \frac{\mathcal{B}[(1+\mathcal{A}L^2n^2\gamma^3)^{t+1}-1]}{\mathcal{A}\gamma n(1-\gamma^2 L^2 n^2)}. \tag{27}$$

Let $\eta \in (1/t, 1)$ (assuming $t > 1$). Setting $\gamma \leq \frac{1}{\sqrt{3}Ln}$, we have $\frac{2\delta_{t+1}}{\gamma n(1-\gamma^2 L^2 n^2)} > 0$. Therefore, the left-hand side in the inequality (27) is bounded from below by

$$\min_{(1-\eta)t \leq j \leq t} r_j \sum_{(1-\eta)t \leq j \leq t} (1 + \mathcal{A}L^2n^2\gamma^3)^{t-j} \geq (\eta t - 1) \min_{(1-\eta)t \leq j \leq t} r_j;$$

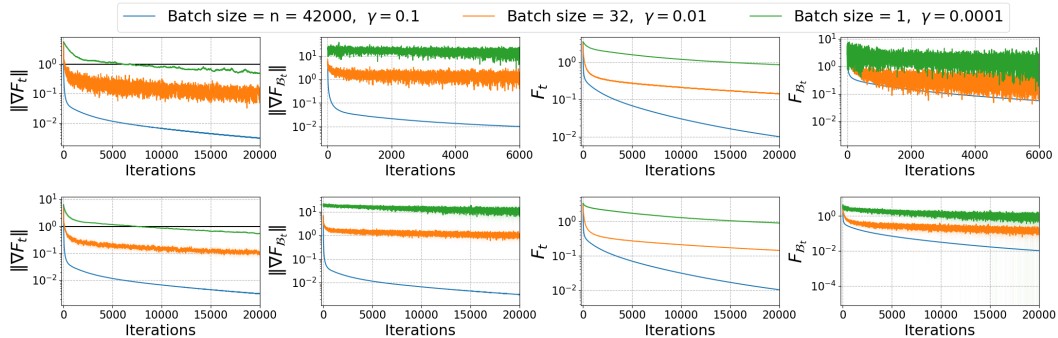

Figure 5: Performance of `RR-SGD` on MNIST digit classification. The top row shows the result of 1 single run of `RR-SGD` while the bottom row shows the result of the average of 10 runs. For the plots in the first column, the horizontal lines correspond to the precision, $\epsilon = 1$—For RR-SGD, if the total number of iterations is large enough then the entire tail comprises of the $\epsilon$-stationary points.

while if $\gamma \leq \frac{1}{\sqrt{3}Ln}$ and $(1 + \mathcal{A}L^2n^2\gamma^3)^{t+1} \leq 3$ then the right-hand side of (27) could be bounded from above by

$$\frac{9\delta_0}{\gamma n} + \frac{3\mathcal{B}}{\mathcal{A}\gamma n}.$$

Hence, we obtain

$$\min_{(1-\eta)t \leq j \leq t} r_j \leq 3\left(3\delta_0 + \frac{\mathcal{B}}{\mathcal{A}}\right)\frac{1}{(\eta t - 1)\gamma n}. \tag{28}$$

Considering the step size

$$\gamma = \gamma_t := \left(\frac{\ln 3}{(t+1)\mathcal{A}L^2n^2}\right)^{\frac{1}{3}},$$

we complete the proof of Theorem 7.

### B.4 CONVERGENCE OF SGD FOR NONCONVEX AND NONSMOOTH OBJECTIVE

Consider a nonconvex, nonsmooth, finite-sum optimization problem by writing (1) of the form:

$$\min_{x \in \mathbb{R}^d}\left[F(x) := \underbrace{\frac{1}{n}\sum_{i=1}^{n}f_i(x)}_{:=f(x)} + h(x)\right], \tag{29}$$

where each $f_i(x) : \mathbb{R}^d \to \mathbb{R}$ is smooth (possibly nonconvex) for all $i \in [n]$, and $h : \mathbb{R}^d \to \mathbb{R}$ is nonsmooth but (non)-convex and relatively simple. Consider a mapping, $\mathcal{G}_\eta : \mathbb{R}^d \to \mathbb{R}$ such that

$$\mathcal{G}_\eta(x) := \frac{1}{\eta}(x - \text{prox}_{\eta h}(x - \nabla f(x))), \tag{30}$$

where for a nonsmooth, proper, closed function, $h$, the proximal operator is defined as

$$\text{prox}_{\eta h}(x) := \arg\min_{y \in \mathbb{R}^d}\left(h(y) + \frac{1}{2\eta}\|y - x\|^2\right). \tag{31}$$

**Theorem 8.** *Under the same assumptions as in Reddi et al. (2016), there exists an index, $t \geq (1-\eta)T$ such that, $\mathbb{E}\|\mathcal{G}_\eta(x_t)\|^2 \leq O\left(\frac{1}{\eta T}\right)$, where $\eta \in (0, 1]$.*

The proof of convergence follows similar argument as in Reddi et al. (2016) combined with our techniques. Moreover, our result can be extended to proximal stochastic gradient algorithms (with or without variance reduction) for nonconvex, nonsmooth finite-sum problems (Reddi et al., 2016; Li & Li, 2018), and for non-convex problems with a non-smooth and non-convex regularizer (Xu et al., 2019).

## C REPRODUCIBLE RESEARCH

Our code and results are publicly available at `https://anonymous.4open.science/r/nonconvex-convergence-SGD-2844/README.md`.