# OpenReview forum: "Demystifying the Myths and Legends of Nonconvex Convergence of SGD"
_ICLR.cc/2024/Conference — ICLR 2024 Conference Withdrawn Submission_

### Official Review · Reviewer_onbN · 2023-10-29

**Soundness:** 1 poor
**Presentation:** 2 fair
**Contribution:** 1 poor
**Rating:** 3
**Confidence:** 4

**Summary:**

This paper studies the convergence guarantees of non-convex SGD in the tail phase of iteration. The main take-away is that for any portion ratio $\eta\in(0,1)$, there always exists some $\epsilon$-stationary point in the final $\eta T$ iterates with sufficiently large iteration count $T$. Additionally, it is claimed that the density of $\epsilon$-stationary points in the last $\eta T$ iterates approaches one as $T$ goes infinity. Numerical results on nonconvex regularized logistic regression and CNN training tasks are reported to confirm the theoretical predictions.

**Strengths:**

- S1: The topic of analyzing the optimality of the final SGD iterates should be of sufficient interests for both machine learning researchers and practitioners, especially in the nonconvex stochastic optimization community.

- S2: Measuring the density of stationary points in the tail phase of SGD iteration is an interesting problem worthy of exploration.

**Weaknesses:**

While the addressed problem is important, the main results obtained in this work are unfortunately not satisfactory both in significance/novelty and correctness.

- W1: Significance of contribution. IMO, one major limitation of the main result in Theorem 3 lies in that the required iteration count $T$ to guarantee the existence of stationary solution in the last $\eta$ portion should scale as $T’/\eta^2$ where $T’$ is the well-known complexity implied by Theorem 1. This makes the result in Theorem 3 much less interesting as the portion size $\eta T$ could be even larger than $T’$, and thus can hard to convincingly demystify the appealing performance of SGD in the tail iteration phase.

- W2: Novelty of analysis. The convergence analysis developed in the current work builds largely upon the existing techniques for nonconvex SGD (e.g., Khaled & Richtarik 2022), with some slight modifications adopted mostly by ignoring the first $(1-\eta)T$ terms in the sum of gradient norms at the iterates. It turns out that such a naïve treatment only leads to fairly weak guarantees as pointed out in the previous comment.

- W3: Soundness of claim. Regarding the results about the density of stationary points, I cannot agree that the bound in Theorem 5 guarantees the entire tail iterates would be stationary when $T$ becomes large enough. The rationale is that the constant step-size is typically chosen as $\gamma=\frac{1}{\sqrt{T}}$ such that $\log(1+L\gamma^2A)$ roughly scales as $L\gamma^2A=\frac{LA}{T}$ which cancels the factor $T$ in the denominator. Concerning Theorem 6, the RHS of the bound looks confusing to me mainly because the role of $\epsilon$ is completely missing in the bound. More specifically, it looks like a factor $1/\epsilon$ is missing in the definition of the quantity $D$ above Theorem 6. Please correct me if anything misunderstood here. Otherwise, the current results in Theorem 5 and Theorem 6 cannot at all support the claim that the final iterates of SGD should be densely concentrated around the $\epsilon$-stationary points.

- W4: Presentation clarity. The readability of this paper can be enhanced by properly highlighting and reorganizing the materials in Section 4. There are all together six theorems stated in Section 4. However, some of these theorems are either restated from literature (Theorem 1), or intermediate results (Theorem 2), or for different choices of learning rates (Theorems 3 - 6). It is suggested to only highlight those relatively most fundamental results as theorems while leave the rest as lemmas or corollaries. Concerning the exposition of experimental results, it is odd to show Figure 1 and Figure 2 in the front sections without any introduction, but quite later get the figures explained in Section 5.

**Questions:**

- Q1 Given that the portion size $\eta T$ would still be large as commented in W1, what are the real advantages of Theorem 3 and Theorem 4, if any, over Theorem 1?

- Q2 Could you more explicilty highlight the technical novelty of this work with respect to the previous analysis for nonconvex SGD?

- Q3 Why the factor $1/\epsilon$ disappears in the quantity $D$ introduced above Theorem 6?

---

### Official Review · Reviewer_9YpN · 2023-10-30

**Soundness:** 2 fair
**Presentation:** 1 poor
**Contribution:** 1 poor
**Rating:** 3
**Confidence:** 4

**Summary:**

This work analyses convergence of non-convex SGD to a stationary point and establishes that the last $100 \eta$ percent of iterates (where $\eta \in (0,1)$) will converge at the rate $O(1/\sqrt{T})$. However, this rate hides the dependence on $\eta$ in the denominator, which significantly weakens the theoretical contribution of the work.

It could be useful to show an example of non-convex function for which such dependence on $\eta$ is tight.

**Strengths:**

The paper approaches an important problem of the last iterate convergence of nonconvex SGD for smooth problems.

**Weaknesses:**

1. Assumption 3. states that constant A can be zero, but throughout the paper there is a division by A. E.g., Corollary 1, Theorem 1, 2. This means that the important classical setting of bounded variance ($A = 0$, $B = 1$, $C = \sigma^2$) cannot be recovered by the theory in this work.

2. The statement at the end of page 1 is wrong. It is well-known and immediate to show the last iterate convergence of SGD under Polyak-Łojasiewicz or even weaker assumptions, see, e.g., Section 5 in [1] or Corollary 1 in [2].

3. The main result of the paper (Theorems 2, 3) seems to be trivial to obtain by simply applying the standard analysis of SGD to the last $\eta T$ iterates for any $\eta \in (0,1)$. Yes, the authors expectedly obtain the optimal convergence rate in $\epsilon$. However, the constant hides the dependence on $\eta$. E.g., taking $\eta = 1/T$ would mean that the last iterate does not even have to converge.

4. The presentation of the myths/legends in the paper seem highly subjective and sometimes confusing. Some of them were already addressed in the literature or believed to be well-known results.
- For instance, Legend I does not seem to be challenged in this paper since no better convergence is established (see point 3.).
- Myth II is not really a myth. It is well known that all what matters for convergence of SGD is the order of the sum of stepsizes and of the sum of its squares. It has been observed in many prior work already.
- Myth III about better assumptions of the noise model is confusing. I do not see how this assumption is challenged/demystified in this work. In fact, even the last iterate convergence of GD (SGD without noise) is not established in this work.


[1] X. Fontaine, V. De Bortoli, A. Durmus. Convergence rates and approximation results for SGD and its continuous-time counterpart. COLT 2021.

[2] I. Fatkhullin, J. Etesami, N. He, N, Kiyavash. Sharp analysis of stochastic optimization under global Kurdyka-Lojasiewicz inequality. NeurIPS 2022.

**Questions:**

-

---

### Official Review · Reviewer_n2mz · 2023-10-31

**Soundness:** 2 fair
**Presentation:** 2 fair
**Contribution:** 1 poor
**Rating:** 3
**Confidence:** 4

**Summary:**

The paper focuses on providing convergence guarantees for the last $\eta$ portion of iterates when we run SGD for nonconvex objectives.

**Strengths:**

1. The paper poses an intriguing research question:

"How is the $\epsilon$-stationary distributed after running SGD for T iterations on a nonconvex problem?"

While the question is compelling, I intuitively feel that it might not have a definitive answer. The distribution can vary significantly depending on the function in question, as elaborated later.

2. The paper adopts a weak assumption on the distribution of the noise.

**Weaknesses:**

1. The guarantees provided for the iterations in the tail are weak or even trivial. Basically, it is based on the argument that: if we only consider the last $\eta T$ iterations, when telescope, we just throw away the first $(1-\eta T)$ terms as we sum up the gradient norms. In other words, we can consider it as two stages, in the first stage of $(1-\eta T)$ iterations, the function value will follow some descent lemma up to an extra term in noise (depending on C and $\gamma_t^2$), i.e., the last term in (3), which is summable; the second stage of last $\eta T$ iterations, it just follow the normal analysis for SGD. Such an argument in Theorem 2, 3, and 4 is not very meaningful because:

a. The results contain $\eta T$ on the right-hand side of (6). This means that the guarantee of the tail $\eta T$ iterations for running T iterations is not better, or even worse, than if we just run $\eta T$ iterations in total. If the dependence on $\eta$ is better than linear, (e.g. $\eta^{0.9} T$) it would be more meaningful.




2. The paper missed some of the most relevant references:

[1] Orabona, F. (2020). Almost sure convergence of sgd on smooth nonconvex functions. Blogpost on http://parameterfree. com, available at https://parameterfree. com/2020/10/05/almost-sure-convergence-of-sgd-on-smooth-non-convex-functions.

[2] Bertsekas, D. P., & Tsitsiklis, J. N. (2000). Gradient convergence in gradient methods with errors. SIAM Journal on Optimization, 10(3), 627-642.

[3] Drori, Y., & Shamir, O. (2020, November). The complexity of finding stationary points with stochastic gradient descent. In International Conference on Machine Learning (pp. 2658-2667). PMLR.

In particular, [1] and [2] show the asymptotic convergence of the last iterate for SGD in nonconvex optimization. [3] shows that for any fixed iterate (e.g., last iterate or k-th iterate) of T iterations of SGD, we can not provide a guarantee for the gradient. These two results do not contradict each other, because [1, 2] are for the asymptotic convergence for a fixed nonconvex problem, while [3] says we cannot provide a non-asymptotic rate for the last-iterate for the class of L-smooth function. The example given in [3] implies that it might be hard to characterize when $\epsilon$-stationary points appear in T iterations for the whole function class.

3. The myths and legends presented on page 7 seem to lack a strong connection with the rest of the paper's content.

**Questions:**

N/A

---

### Official Review · Reviewer_5pB8 · 2023-11-08

**Soundness:** 1 poor
**Presentation:** 1 poor
**Contribution:** 1 poor
**Rating:** 1
**Confidence:** 4

**Summary:**

This paper considers the last iteration complexity bound of SGD for smooth nonconvex optimization. They obtain such a bound for the final iterates.

**Strengths:**

It is relatively easy to review this paper.

**Weaknesses:**

The result is trivial.

This paper does not really provide the last iterate complexity bound for nonconvex SGD. Instead, it is still an averaged or minimal gradient norm complexity result, but with respect to the last $\eta T$ iterations for some **constant $\eta$**.

We can take Theorem 3 as an example. The proof is to apply the traditional SGD analysis to the last $\eta T$ iterations, meaning that the result is trivial. Let us **fix the total number of iterations to be $\eta T$ rather than $T$**. Through a set of quite standard analyses of nonconvex SGD, we have $\min_{0\leq t \eta T} \mathbb{E}\|\nabla f(x_t\|^2\leq \varepsilon$, if $\eta T \geq \mathcal{O}(1/\varepsilon^2)$.

**Questions:**

See the weakness above.